# *MYD88* Mutations: Transforming the Landscape of IgM Monoclonal Gammopathies

**DOI:** 10.3390/ijms23105570

**Published:** 2022-05-16

**Authors:** Miguel Alcoceba, María García-Álvarez, Alejandro Medina, Rebeca Maldonado, Verónica González-Calle, María Carmen Chillón, María Eugenia Sarasquete, Marcos González, Ramón García-Sanz, Cristina Jiménez

**Affiliations:** Hematology Department, University Hospital of Salamanca (HUS/IBSAL), CIBERONC and Cancer Research Institute of Salamanca-IBMCC (USAL-CSIC), 37007 Salamanca, Spain; alcocebasanchez@saludcastillayleon.es (M.A.); mgarca1991@gmail.com (M.G.-Á.); amedinah@saludcastillayleon.es (A.M.); rebecamaldonadosanchez@gmail.com (R.M.); vgcalle@saludcastillayleon.es (V.G.-C.); mcchillon@saludcastillayleon.es (M.C.C.); mealonsos@saludcastillayleon.es (M.E.S.); marcosgonzalez@saludcastillayleon.es (M.G.); jscris@usal.es (C.J.)

**Keywords:** signal transduction, hematologic neoplasms, targeted therapy, Waldenström’s macroglobulinemia

## Abstract

The *MYD88* gene has a physiological role in the innate immune system. Somatic mutations in *MYD88*, including the most common L265P, have been associated with the development of certain types of lymphoma. *MYD88^L265P^* is present in more than 90% of patients with Waldenström’s macroglobulinemia (WM) and IgM monoclonal gammopathy of undetermined significance (IgM-MGUS). The absence of *MYD88* mutations in WM patients has been associated with a higher risk of transformation into aggressive lymphoma, resistance to certain therapies (BTK inhibitors), and shorter overall survival. The MyD88 signaling pathway has also been used as a target for specific therapies. In this review, we summarize the clinical applications of *MYD88* testing in the diagnosis, prognosis, follow-up, and treatment of patients. Although *MYD88^L265P^* is not specific to WM, few tumors present a single causative mutation in a recurrent position. The role of the oncogene in the pathogenesis of WM is still unclear, especially considering that the mutation can be found in normal B cells of patients, as recently reported. This may have important implications for early lymphoma detection in healthy elderly individuals and for the treatment response assessment based on a *MYD88^L265P^* analysis.

## 1. Introduction

The myeloid differentiation primary response 88 (*MYD88*) gene encodes a cytosolic adaptor protein that plays a central role in the innate and adaptive immune responses mediated by the interleukin (IL) 1 receptor (IL-1R) family and toll-like receptors (TLRs) [1]. The MyD88 protein has a modular structure composed of three main domains: an N-terminal death domain (DD), responsible for binding to the IL-1R-associated kinase (IRAK) complex and for further signaling through the pathway [2], a toll-interleukin-1 receptor (TIR) domain, located at the C-terminus and that binds to the receptors [3], and an intermediary domain (INT), which separates both [1]. Based on its unique structure, MyD88 serves as the central link that connects TLR-IL-1R family members to IRAKs (IRAK1, IRAK2, and IRAK4) [4,5], leading to the activation of the nuclear factor kappa B (NF-κB), mitogen-activated protein kinase (MAPK), and interferon regulatory factor (IRF) 5/7 signaling pathways and the downstream production of type I interferons and proinflammatory cytokines, including IL-1, IL-6, IL-12, and tumor necrosis factor (TNF)-α [2,6,7,8,9,10].

In addition to its physiological role in the immune response, *MYD88* can also act as an oncogene associated with lymphoma development, and somatic mutations have been identified in numerous patients. The most recurrent, a single nucleotide change from T to C, resulting in a change from leucine to proline at position 265 (L265P) (Figure 1), is present in more than 90% of Waldenström’s macroglobulinemia (WM) patients [11,12,13,14,15], in whom it has been shown to have important clinical and therapeutic implications [16]. However, its role in the disease pathogenesis has not yet been fully elucidated. In this paper, we will provide a summary view of the gene function in normal and pathological conditions, the clinical applications, including diagnosis and therapy, and will comment on the latest findings about *MYD88^L265P^* not being exclusive to tumor cells and the potential implications derived from this.

## 2. Signaling Pathway and Molecular Alterations

In normal physiology, MyD88 is required for full function of the innate immune response as a signaling adaptor in the canonical NF-κB pathway [17]. Upon ligand binding, MyD88 dimerizes, through its death domain (DD), and recruits IRAK4, followed by IRAK1/2, resulting in the assembly of a large complex called ‘Myddosome’, which consists of six MyD88 molecules, four IRAK4 molecules, and four IRAK1/2 molecules [8,18,19]. IRAK1/2 can subsequently interact with TNF receptor-associated factor 6 (TRAF6) to activate transforming growth factor beta-activated kinase 1 (TAK1), which initiates a phosphorylation cascade, resulting in the release of the NF-κB subunits RELA [p65]-p50 and RELB-p52, which migrate to the nucleus, bind to the DNA, and increase the expression of the target genes [19].

In the context of cancer and, particularly, in hematology, one can hardly speak about MyD88 without thinking of WM. WM is an indolent hematological malignancy that makes up approximately 1% of non-Hodgkin’s lymphomas (NHL). Patients present both abnormal cells in their bone marrow biopsy and abnormal IgM macroglobulin in their blood [20]. Such as other low-grade lymphoproliferative malignancies, WM is an incurable disease, although with a variable clinical course. Thus, there are patients who remain asymptomatic for 5–10 years, while high-risk patients have a median survival of around 3 years [21,22]. Until 2012, very little was known about the genomics underlying the disease. The discovery of the gain-of-function *MYD88^L265P^* mutation was a turning point for WM, providing valuable insights into the signaling pathways involved in this malignancy [11].

The L265P mutation makes the TIR domain of MyD88 more activate compared to the wild-type protein, which increases the formation of the ‘Myddosome’ complex and the downstream signaling [23,24,25]. Thus, the functional effects of *MYD88^L265P^* include increased NF-κB activity; Janus kinase (JAK)/signal transducer and activator of transcription (STAT) signaling; and the production of proinflammatory cytokines, such as IL-6, IL-10 and interferon (IFN)-β, as well as the enhanced growth and survival of lymphoma cells [23,26].

In addition to the NF-κB pathway, the B-cell receptor (BCR) pathway also plays an important role in the oncogenesis of B-NHL with *MYD88* mutations. In normal conditions, the BCR signaling activates NF-κB, phosphoinositide 3-kinase (PI3K), MAPK, and nuclear factor of activated T cells (NFAT) pathways. Bruton’s tyrosine kinase (BTK), a protein of the BCR signaling cascade, has been shown to preferentially form complexes with mutated MyD88 and not with wild-type MyD88 in WM cells. *MYD88^L265P^* triggers BTK and downstream NF-kB signaling independent of IRAK1 and -4 [26]. In addition, *MYD88^L265P^* is common in patients with mutations in BCR-associated protein CD79B, and these patients benefit most from the treatment with BTK inhibitors [27,28,29]. The mechanism of combined MyD88 and BCR pathway activation may be explained by the existence of a ‘MyD88-TLR9-BCR (My-T-BCR) supercomplex’ that encompasses mutated MyD88 and BCR components and contributes to broader signaling, including the mammalian target of rapamycin (mTOR) and NF-κB pathways, thereby promoting lymphomagenesis [30].

There is also evidence of cross-communication with the BCR pathway triggered by mutated MyD88 through the activation of the BCR-signaling component SYK (spleen tyrosine kinase) in WM and activated B-cell diffuse large B-cell lymphoma (ABC DLBCL) cells [31]. SYK can also be activated by the hematopoietic cell kinase (HCK) in MyD88-mutated lymphoma cells [32], and, at the same time, HCK can be triggered either by mutated MyD88 directly or through IL-6. For its part, the HCK protein promotes lymphomagenesis via multiple signaling pathways, including BTK, PI3K, and MAPK [33].

To complete the picture, NF-κB activity not only activates B-cell proliferation and survival-related genes but also results in IL-6 and Il-10 autocrine signaling, which, via JAK/STAT or HCK, increases the transcription of genes involved in several signaling cascades, such as PI3K/AKT/mTOR, JAK/STAT, or NF-κB [34].

The *MYD88* alteration has been found at high frequencies in cutaneous DLBCL (69%), primary testicular lymphoma (68%), primary central nervous system lymphoma (38–86%), or ABC DLBCL (30%), indicating its role in the pathogenesis of lymphoid neoplasia [23,35,36,37,38,39]. In WM, a small number of patients (7%) lack *MYD88* mutations. These patients present a different genomic landscape, including other NF-κB-activating mutations, epigenomic dysregulation, and impaired DNA damage repair, which leads to inferior outcomes compared to those with MyD88 mutations—specifically, a shorter overall survival (OS) and a higher risk of histological transformation [40,41,42,43]. These findings have prompted some authors to consider the disease with the *MYD88* wild-type genotype to be an entirely separate entity, proposing the presence of the MyD88 mutation as a WM-defining feature [44,45]. Few cancers have a single amino acid substitution in one gene present in most cases, making WM paradigmatic for studying the role of a single causative mutation in oncogenesis. The presence of *MYD88^L265P^* in IgM-MGUS suggests that it is an early oncogenic factor, but most IgM-MGUS patients never progress to WM or other lymphoproliferative disorders, so this mutation cannot be considered a unique pathogenic factor in WM [46,47]. This is consistent with animal models, where *MYD88* mutations are not sufficient for the development of WM or other MyD88-driven lymphomas, and additional alterations in tumor cells and/or the host response are needed [48,49].

## 3. Clinical Applications

The *MYD88^L265P^* mutation has become the hallmark of WM, as it is present in more than 90% of patients. Therefore, numerous clinical applications have emerged concerning its diagnosis, management, and treatment.

### 3.1. Diagnosis

Currently, the diagnosis of WM is contingent on demonstrating a lymphoplasmacytic cell infiltrate, usually by a bone marrow biopsy, which has several disadvantages, such as patient discomfort, unforeseen complications, high cost, and delay in the diagnosis. In contrast to WM and IgM-MGUS (>90%) [12,13,14,15,46], *MYD88^L265P^* is absent or less frequent in other related B-cell (including IgM-secreting) disorders, such as mucosa-associated lymphoid tissue lymphoma, splenic marginal zone lymphoma, nodal marginal zone lymphoma, IgM-secreting multiple myeloma, and chronic lymphocytic leukemia, therefore representing a useful tool for discriminating WM from other related B-cell disorders at diagnosis (Table 1) [23,24,50,51,52,53,54,55,56]. In addition, several studies support the feasibility of using allele-specific polymerase chain reaction (AS-PCR), digital PCR (dPCR), and next-generation sequencing assays for *MYD88^L265P^* detection in peripheral blood and circulating tumor DNA, thereby providing convenient and less invasive methods for the diagnosis of WM and IgM-MGUS [57,58,59,60,61], similar to other mutations associated with hematological conditions, e.g., chronic myeloid leukemia (BCR-ABL), polycythemia vera (*JAK2^V617F^*), and hairy cell leukemia (*BRAF^V600E^*) [62,63,64].

*MYD88^L265P^* detection in the cerebrospinal fluid by dPCR is also useful to diagnose Bing-Neel syndrome and central nervous system (CNS) lymphomas as an alternative to a cerebral or retinal biopsy [65,66], since, in clinical practice, the collection of tumor tissue is a highly invasive procedure hampered by the risk of severe complications. Remarkably, the MyD88 mutation never occurs in tissue biopsies from nonhematologic brain tumors, such as a glioblastoma, or in solid metastatic tumors, suggesting it is a sensitive and specific biomarker for the differentiation of primary central nervous system lymphoma from other CNS cancers [67]. Furthermore, the high frequency of *MYD88^L265P^* in these diseases makes this mutation a perfect candidate for liquid biopsy to enter clinical practice [35].

### 3.2. Follow-Up

Although minimal residual disease (MRD) monitoring in certain lymphomas, such as follicular lymphoma, has been gradually established, it has only started to be explored in WM [68]. Several studies have shown the role of *MYD88^L265P^* as a predictive biomarker of the therapy response in the bone marrow and peripheral blood compartments [13,15,57]. The application of *MYD88^L265P^* testing has been demonstrated to be more useful than serum IgM in estimating the underlying disease burden, especially with agents affecting the serum IgM levels, either by inducing an IgM flare or by blocking IgM secretion, such as rituximab, bortezomib, everolimus, and ibrutinib [69]. Increased serum IgM can be mistaken for disease progression, leading to a drug change, whereas a blockage in IgM secretion out of proportion to the tumor load lends to underestimating the posttreatment disease burden, missing the disease progression in some instances. Therefore, the use of peripheral blood *MYD88^L265P^* testing to estimate the underlying disease burden in patients undergoing those treatments could help guide the clinical management and avoid the repetition of bone marrow biopsies to clarify IgM discordance [57]. Nevertheless, the value of using *MYD88^L265P^* in the response assessment still needs to be validated in a larger series of treated WM patients and, ideally, across multiple therapeutic regimens. In addition to peripheral blood samples, circulating tumor DNA may represent another attractive, noninvasive alternative to bone marrow [68].

### 3.3. Prognosis

Data regarding the prognostic value of MyD88 mutation are quite controversial. The initial studies showed that *MYD88^L265P^* was associated with a higher risk of disease progression in IgM-MGUS and asymptomatic WM [11,12,13,70,71]. However, other authors have identified the absence of the alteration as an independent risk factor for progression to WM or other lymphoproliferative disorders, although, in some studies, this observation did not reach statistical significance, likely as a reflection of the small sample size [72,73]. There is also another perspective that considers that those patients who progressed were precursors of WM rather than transformations from IgM-MGUS to WM, thus suggesting that the acquisition of *MYD88^L265P^* would not represent a transformation event [46]. Therefore, further studies are needed to clear up this matter. An assessment of the *MYD88* status should be included in the initial work-up of all patients with IgM-MGUS/asymptomatic WM to help clarify whether mutated patients show a higher risk of evolution to WM. Furthermore, patients with wild-type *MYD88* should be followed closely because of their higher risk of histological transformation and development of therapy-related myelodysplastic syndrome [41,42,43,44,74].

The impact of *MYD88* mutation on the disease outcome is also unclear. Although it is fairly well-established that the wild-type genotype exhibits an increased risk of death (estimated 10-year survival is 73% for *MYD88^wild-type^* vs. 90% for mutated *MYD88*) [40,75], other groups have observed similar OS and similar times to the next treatment after the frontline therapy in *MYD88^L265P^* and *MYD88^wild-type^* patient populations [44,76].

The prognostic significance of the *MYD88* status in other diseases has not been fully elucidated either. This is very well-exemplified by CLL, where studies have shown strong contradictions, both in the frequency of *MYD88* mutations (from 1.5 to 10% of patients) [24,77] and in the prognostic significance: from favorable [78] to neutral [79] or unfavorable [80]. In ABC DLBCL, a negative effect of *MYD88^L265P^* on patient survival seems evident, but it is still considered a matter of debate [39,81,82,83].

### 3.4. Treatment

There are multiple treatment options for WM patients, such as chemotherapy, monoclonal antibodies, proteasome inhibitors, and BTK inhibitors. The choice for therapy should consider the clinical presentation, comorbidities, and preferences. However, there is increasing evidence that the genomic profile may provide insightful information for treatment selection. Thus, the detection of *MYD88^L265P^* may help identify those patients who are more suitable for treatments targeting the MyD88-driven pathways, as described below.

Thanks to the discovery of the MyD88 mutation in WM, BTK inhibitors have become an important treatment option. Much of their efficacy is due to the presence of this alteration, which has been shown to serve as a predictor of the response in patients treated with a BTK inhibitor-based regimen. In previously treated WM, the major response rate to ibrutinib therapy was found to be substantially higher for *MYD88^L265P^-CXCR4^wild-type^* (97%) and *MYD88^L265P^-CXCR4^WHIM^* (68%) compared to patients with the *MYD88* wild-type genotype (0%) [43,84,85]. In treatment-naïve patients, the overall response rate and major response rate were, respectively, 100% and 94% for *MYD88^L265P^-CXCR4^wild-type^* patients [86]. Nevertheless, another study observed no influence of the genotype in the response rate and time to respond when comparing ibrutinib–rituximab vs. placebo–rituximab [87]. Zanubrutinib, a second-generation BTK inhibitor, has demonstrated high-quality responses in *MYD88^wild-type^* WM patients, including 27% of the very good partial responses and 50% of the major responses [88]. The overall response rate was similar regardless of the genotype. However, the proportions of a very good partial response and complete response were still different, since they were higher for *MYD88^L265P^-CXCR4^WHIM^* patients (59%) compared to *MYD88^wild-type^* (25%) [89]. Acalabrutinib, another second-generation BTK inhibitor, also provides a good response rate in WM, although no sufficient data are yet known about the potential influence of genomics [90].

The National Comprehensive Cancer Network^®^ Guidelines for WM (version 2.2022) include *MYD88^L265P^* testing of the bone marrow and a genomic-based treatment approach to symptomatic treatment-naïve and relapsed or refractory WM [91]. Concordantly, the last international workshop on a WM consensus panel recommended against the use of ibrutinib monotherapy in *MYD88^wild-type^* patients [16].

In ABC DLBCL, the *MYD88/CD79B* double mutant shows an intense sensitivity to BTK inhibitors, while patients with mutated MyD88 but without BCR mutations (i.e., CD79A or CD79B) are not responsive [28,92]. The molecular basis of this observation might be the induction by abnormal MyD88 of a chronically active BCR signaling through the formation of the ‘My-T-BCR supercomplex’, thus mitigating its sensitivity to ibrutinib [30]. In WM, alterations concerning the key negative regulators of BTK, MyD88, and NF-κB, as well as the ubiquitin ligase and TLR pathway regulators, are involved in ibrutinib resistance [93,94].

### 3.5. Therapeutic Target

*MYD88^L265P^* is also significant given the interest in therapies targeting the components of pathways activated by this mutation (Figure 2). Therapeutic targets that have been, or are currently being, investigated include BTK in the BCR pathway (as already reviewed); TLRs, IL-1R, and their ligands; IRAK1 and IRAK4 in the ‘Myddosome’ complex; TAK1 in downstream signaling; and components of the HCK and PI3K/AKT/mTOR pathways [95,96,97,98,99,100,101].

The blockade of TLRs ligand activation results in cell signaling inhibition, tumor growth reduction, and the induction of apoptosis in *MYD88^L265P^* WM cells. Oligonucleotides that inhibit TLR7/8/9 have already been tested in clinical trials for *MYD88*-mutated DLBCL and WM patients [102,103]. Nevertheless, it is possible that L265P induces NF-κB activation completely independently of any upstream receptor, so that therapeutic options targeting the downstream pathway may be mechanistically more effective.

MyD88 is involved in the IRAK-mediated activation of TLR signaling. The initial IRAK inhibitors showed promising results, blocking IRAK4 in vitro and in xenografts with human DLBCL cell lines [95,97]. Recent studies have demonstrated the potential superior antitumor activity of these compounds in combination with ibrutinib, bortezomib, or venetoclax in preclinical models of WM, *MYD88*-mutated DLBCL, and CLL [104,105,106,107,108]. However, IRAKs are not involved in all MyD88-dependent signaling, so that MyD88 dimerization may represent a more favorable target. Mini-peptides that compete with MyD88 TIR domain interactions prevent MyD88 dimerization and ‘Myddosome’ signaling, leading to the inhibition of NF-κB activity and cell survival [109,110,111]. L265P-mutant cell lines have been shown to be much more sensitive to this inhibition than wild-type cells [112].

HCK activation is triggered by mutated MyD88 and promotes malignant cell growth and survival through BTK. KIN-8194 is a novel dual inhibitor of HCK and BTK that has shown the potent and selective in vitro killing of *MYD88*-mutated lymphoma cells, including ibrutinib-resistant *BTK^C481S^*-expressing cells, and demonstrated excellent bioavailability, pharmacokinetic parameters, and good tolerance at active doses [113]. Pirtobrutinib, a third-generation noncovalent BTK inhibitor, has a different binding site in BTK, being able to overcome the resistance associated with mutations at C481. Pirtobrutinib inhibits growth and can trigger the apoptosis of MyD88-mutated lymphoma cells in a highly selective manner, thus improving their efficacy and tolerance [114,115].

A novel personalized therapeutic strategy are *MYD88^L265P^*-derived peptides. These tumor-specific neoepitopes are identified as foreign by the immune system, inducing tumor-specific T-cell immunity. It has been demonstrated that healthy individuals harbor T cells with specific T-cell receptors (TCRs) that are able to recognize L265P-containing neoantigens and elicit human leukocyte antigen (HLA) class I-restricted cytotoxic T-cell responses (when presented by HLA-B*07 and -B*15), supporting the potential for TCR-based immunotherapy. The peptide-specific cytolytic activity of CD8+ T cells was not observed when wild-type MyD88 peptides were presented [116,117]. Nevertheless, further studies are needed to better understand the functional properties of *MYD88^L265P^*-specific T cells in the tumor niche, identify specific major histocompatibility complex haplotypes for TCRs that could be used for immunotherapy, and quantify the extent to which mutant MyD88 is a target of specific immunity [118,119].

Finally, chimeric antigen receptor (CAR) T cells have provided a new opportunity to specifically exploit T-cell-intrinsic TLR functions, i.e., activating TLR signaling in tumor-recognizing T cells [120,121,122,123]. This can be achieved by expressing TLR signaling or MyD88 domains within or alongside the CAR. CAR-modified T-cell therapy using a second-generation CAR derived from a CD19-directed antibody fused to the ζ chain of CD3 and the intracellular signaling domain of CD28 (19-28z) has shown robust preclinical activity against WM cells [124]. In another study, MyD88 was employed alongside CD40 in an inducible costimulatory complex consisting of a chemical inducer of the dimerization-binding domain and co-expressed with a first-generation CAR construct in T cells [125]. These inducible MyD88/CD40 CAR T cells exhibited superior T-cell proliferation, cytokine production, and tumor killing ability compared to second-generation CAR T cells that did not contain the inducible MyD88/CD40 molecule. CD19- and CD123-targeting MyD88/CD40 CAR T cells have also been tested [122]. In addition to CARs, MyD88 domains are being successfully used in other synthetic T-cell-stimulatory molecules. CD8α:MyD88, a synthetic coreceptor that joins the extracellular and transmembrane domains of CD8α and the intermediate and death domains of MyD88, is able to activate the TLR-signaling pathway in T cells. The CD8α portion interacts with the TCR, leading to TLR pathway activation through the fused MyD88 intracellular domain and resulting in increased effector function and decreased T-cell exhaustion [126].

## 4. Current State-of-the-Art of MyD88

The *MYD88^L265P^* mutation is a disease-defining genetic alteration of WM (95–97%) and IgM-MGUS (90%) that can be used for the diagnosis and monitoring of the disease [11,12,13,14,15]. Despite being a unifying event, its role in the disease pathogenesis is not entirely clear. Its presence in the precursor condition (IgM-MGUS) suggests that is a ‘driver mutation’ or tumor-initiating event, which might provide the early tumor clone a competitive growth advantage and predispose it toward further genetic alterations, since it is insufficient by itself for the full development of lymphomas [48,49]. It has also been considered a progression event, although no other potential driver events (i.e., highly prevalent mutations or genetic abnormalities) have been identified [11].

Recent works have provided interesting and unexpected findings about the mutation. *MYD88^L265P^* was shown to be present in B-cell precursors in 7/10 patients and in residual normal B cells in 6/6 patients [127]. Another study also reported that *MYD88* mutations were present in pre-B progenitors, being detected in more than 20% of the progenitor cells, and in the nonclonal B cells of WM patients [118]. These data demonstrate that the *MYD88* mutation can occur early in lymphopoiesis in WM, before the expansion of the malignant B-cell clone, and that it is not restricted to the clonal population. They also reinforce the idea that *MYD88^L265P^* alone does not conduce a malignant transformation. Other genetic changes (e.g., del(6q), *CXCR4*, *CD79B*, *ARID1A*, *TNFAIP3*, *TP53*, and *BCL2*) are required to cooperate with *MYD88^L265P^,* providing an advantage for B-cell clonal selection and leading to the different types of B-cell malignancies [27,127,128]. In fact, most somatic mutations detected in progenitor cells are undetectable in mature B lymphocytes, suggesting continuous B-cell clonal selection during lymphopoiesis. However, since not all of the tumor cells harbor other genetic changes, it is unknown what makes the difference between a normal cell and a tumor cell when both are MyD88-mutated or, in other words, what makes ‘normal’ cells acquire the tumor phenotype. It has been shown that the acquisition of the *MYD88* mutation in hematopoietic progenitors is associated with changes in the immune microenvironment that lead to progressive growth and evolution of the B-cell clone [118]. In addition, studies have suggested that B cells harboring the *MYD88^L265P^* mutation still require TLR signals to maintain their proliferation and continued survival [30,129,130]. Therefore, it could be hypothesized that lymphomagenesis is driven by the interaction of microbial or viral ligands with the TLR and the subsequent changes in the upstream pathway of MyD88-mutated lymphomas, which may be different in tumor cells and ‘normal’ cells. Moreover, as not all mutated IgM-MGUS progress to the symptomatic disease, it must be discussed whether *MYD88^L265P^* should always be considered a preneoplastic event and, therefore, a biomarker for the early detection of B-cell lymphomas. The presence of mutations in earlier progenitors opens up new lines of investigation, which should be extended to other *MYD88*-mutated lymphomas.

Another important consequence derived from these findings is whether the presence of *MYD88^L265P^* in ‘normal’ cells of patients could lead to false-positive results in MRD monitoring. An analysis of *MYD88* has shown a limited prognostic value for evaluating the treatment efficacy when compared to multiparametric flow cytometry (MFC) [127]. As *MYD88^L265P^* is present in phenotypically normal B cells, a positive PCR result does not imply the persistence of clonal tumor cells in the B-cell compartment. Nevertheless, further studies are required to confirm these results, increasing the sensitivity of MFC to ensure that real residual clonal cells are detected and assessing both B-lymphocyte and plasma cell populations [131,132,133].

## 5. Summary

*MYD88* is an important oncogene that can be recurrently mutated in several types of lymphoma. An assessment of the *MYD88^L265P^* alteration has been shown to improve lymphoma diagnosis and treatment and is becoming increasingly requested for certain entities, such as IgM monoclonal disorders, among all B-cell lymphoproliferative disorders. Its role in disease pathogenesis is not yet well-established, since recent findings have reported the presence of the mutation in earlier progenitors and mature B lymphocytes, suggesting that further alterations or changes in the tumor immune microenvironment are needed to drive the oncogenic transformation. Finally, MyD88-mutant progenitors may be an interesting target for therapy to improve WM curability.

## Figures and Tables

**Figure 1 ijms-23-05570-f001:**
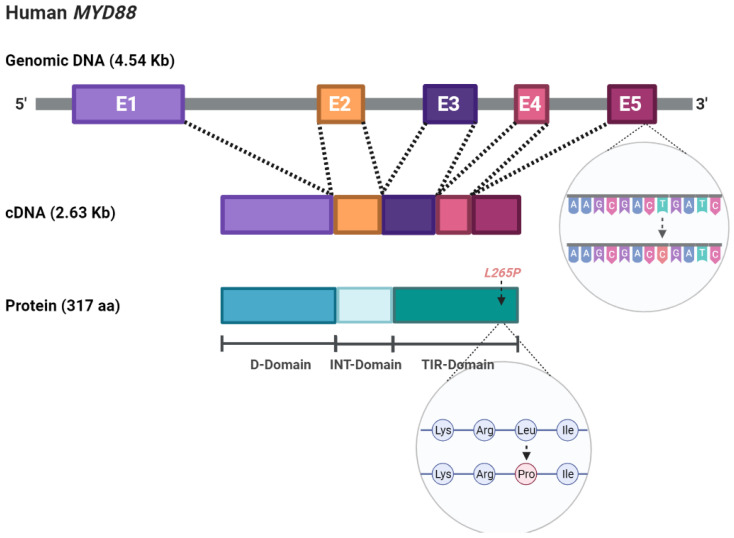
Schematic representation of the *MYD88* gene structure and the L265P mutation at the DNA and protein levels.

**Figure 2 ijms-23-05570-f002:**
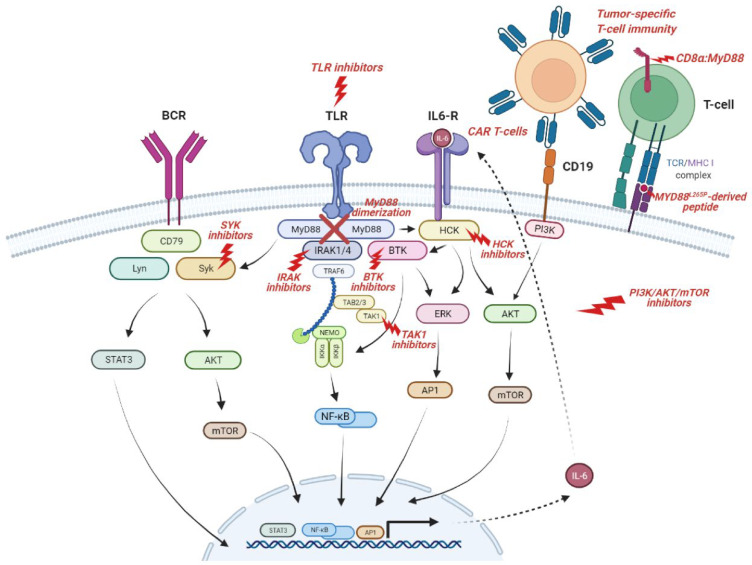
Representation of the MyD88 and related pathways that can be targeted by specific drugs. Components of the pathways activated by *MYD88^L265P^* mutation, such as BTK, IRAKs, and HCK, have proven to be relevant targets for lymphomas. MyD88 is also being used for T-cell immunotherapy as part of CAR T cells, synthetic coreceptors (CD8α:MyD88), or peptides that can induce T-cell responses.

**Table 1 ijms-23-05570-t001:** The frequency of *MYD88^L265P^* in B-cell lymphoproliferative disorders.

Entity	N	*MYD88^L265P^* Range	References
Waldenström’s macroglobulinemia	470	67–100%	[11,12,13,14,15,50,51,52,56]
IgM-MGUS	164	10–87%	[11,12,13,14,15,46]
MALT lymphoma	105	0–9%	[23,50,54]
MZL	325	0–21%	[11,12,13,14,15,50,53,54]
Multiple myeloma (including IgM)	188	0%	[11,13,14,15,51,53,55,56]
Chronic lymphocytic leukemia	412	0–43% ^1^	[13,14,15,24,52,53,56]

^1^ Forty-three percent in a series of CLL with an IgM component. IgM-MGUS, IgM monoclonal gammopathy of undetermined significance; MALT, mucosa-associated lymphoid tissue; MZL, marginal zone lymphoma.

## Data Availability

Not applicable.

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
