# Peer review of "MYD88 Mutations: Transforming the Landscape of IgM Monoclonal Gammopathies"

_ijms, 2022, doi:10.3390/ijms23105570_

Round 1
Reviewer 1 Report
In the review "MYD88 mutations: transforming the landscape of IgM mono-2 clonal gammopathies", Alcoceba M. et al., comprehensively describe the types of lymphoma specifically associated with somatic mutations of MYD88gene in the innate immune system of patients and outline the clinical applications.
This review is well written and provides an in-depth summary of the gene's molecular function, details regarding the cascade of biological signaling pathway in normal and pathological conditions and its prognostic clinical value at early detection of Waldenström’s macroglobulinemia and IgM-MGUS and recent developments in the treatment options including immunotherapy.
They also address the directions to study the limitations in the field, especially the MYD88 mutations involving 'normal cells' lowering the sensitivity and also other additional alterations in the tumor cells.
This review will help understand the state of the art and advance knowledge about different B-cell malignancy detection and therapy.
I have a few minor comments-
1. Are there any information authors can provide or comment on the different TLR ligands(or their sources as we know the microbial and viral components as ligands to TLR signaling ) and subsequent change in their role in the upstream of mutated MYD88 involved in aberrant signaling specific to tumor cells, than that occurs in mutated 'normal cells'?
2. The resolution of the figure 2 maybe improved.
Reviewer 2 Report
In this review the authors provide a summary view of the MYD88 gene function, in normal and pathological conditions, along with the clinical applications, including diagnosis and therapy, of MYD88 mutations. The manuscript is well-written and provides an interesting overview of the topic.
1) Page 4, line 132; Please restate "MYD88L265P is absent or rarely expressed" since it can be found in up to 20% of patients with MZL
2)Page 5, lines 218 - 219; Please check the sentence for syntax
3) Can the authors comment on how reliable are the results of the MYD88 assessment on the peripheral blood in comparison to the bone marrow? Is it safe to omit the cell selection step?
